# OpenReview forum: "Retaining by Doing: The Role of On-Policy Data in Mitigating Forgetting"
_ICML.cc/2026/Conference — ICML 2026 regular_

### Official Review · Reviewer_Dd4F · 2026-03-08

**Soundness:** 3
**Presentation:** 2
**Significance:** 2
**Originality:** 3
**Overall Recommendation:** 4
**Confidence:** 3

**Summary:**

The paper investigates catastrophic forgetting in language-model post-training and shows that, across Llama and Qwen models and tasks like instruction following, MMLU, and arithmetic, RL reliably achieves similar or better target-task performance with much less degradation of other abilities than SFT. By analyzing SFT as forward KL (mode-covering) and RL as reverse KL (mode-seeking) in mixture-of-Gaussians toy setups, the authors find that with realistic multi-modal policies, reverse KL can preserve old modes better than forward KL.

Extensive ablations indicate that this robustness comes mainly from training on on-policy data, not from KL regularization or specific advantage estimators. They further demonstrate that approximately on-policy SFT, e.g., iterative self-training where data is regenerated each epoch, or SFT on traces collected during RL, substantially reduces forgetting at lower compute cost.

**Compliance With Llm Reviewing Policy:**

Affirmed.

**Final Justification:**

I have no further questions and am inclined to recommend acceptance.

However, I find the contributions somewhat straightforward and do not see particularly exciting new findings, so I would also understand if the AC and SAC ultimately decide on rejection.

**Key Questions For Authors:**

Please see the Pros and Cons

**Limitations:**

Please see the Pros and Cons

**Strengths And Weaknesses:**

The authors state that the comparison between SFT and RL in terms of their susceptibility to forgetting is limited, yet there are many prior works, for example, [1-2]. So, I think the authors may a little bit of overclaim of their contribution. Some statements like understanding why RL is less susceptible to forgetting from a new perspective seems to be more reasonable, as no prior works state that using on policy data mainly contributes to the superiority of RL.

[1] RL Is Neither a Panacea Nor a Mirage: Understanding Supervised vs. Reinforcement Learning Fine-Tuning for LLMs

[2]  Mitigating Forgetting Between Supervised and Reinforcement Learning Yields Stronger Reasoner

The main contribution of this paper is that using on policy data is the main reason for the superiority of RL over SFT. Yet, the authors spent a lot of space to show the difference between forward and inverse KL is not the main cause. From my view, this part of the discussion contributes a little to the conclusion and algorithmic improvement, so I think the paper is not that well-structured. It will be great if the authors can clarify this point.

I read some recent papers which state that RL can be viewed as positive + negative learning. So, I view the algorithmic improvement in Sec 4 as some coarse-grained RL method with only positive learning. I think the contribution of this paper is not that enough. I am also wondering in what exact practical cases we need to use this new method.

From Figure 7, it seems that the new method coverages slower than original SFT. I may misunderstand the figure, because the training curves of original SFT are not provided.

---

> ### Author Rebuttal · Authors · 2026-03-31
>
> Thank you for your time and feedback. We treat your comments and questions below. If our response addresses your concerns, we would greatly appreciate it if you would consider raising your score. Please let us know if you have any further questions; we will gladly respond.
>
> **The authors state that the comparison between SFT and RL in terms of their susceptibility to forgetting is limited, yet there are many prior works, for example, [1-2]. So, I think the authors may a little bit of overclaim of their contribution.**
>
> We thank the reviewer for pointing us to references [1] and [2], and we will make sure to cite and discuss them in the revision. We note that:
>
> - Reference [1] compares SFT and RL for reasoning, but its focus is on understanding when RL outperforms SFT, not on systematically identifying why RL forgets less.
> - Reference [2] proposes a method to mitigate forgetting between SFT and RL stages but does not isolate the mechanism behind RL's robustness.
>
> Our core contribution is not merely observing that RL forgets less (which has indeed been noted), but identifying on-policy data as the causal mechanism and ruling out alternative explanations (KL regularization, advantage estimation). This is a distinct and actionable insight. We will revise the introduction to more carefully scope our novelty claims, emphasizing the mechanistic identification over the empirical observation.
>
> **Clarifying the role of the forward and inverse KL analysis.**
>
> > Yet, the authors spent a lot of space to show the difference between forward and inverse KL is not the main cause. From my view, this part of the discussion contributes a little to the conclusion and algorithmic improvement, so I think the paper is not that well-structured.
>
> Thank you for the helpful comment. We believe that there may have been a misunderstanding of what the analysis in Section 3 implies. As detailed below, we will clarify its role in the manuscript.
>
> The analysis serves two important purposes. First, it resolves a conflict between conventional wisdom, which predicts mode-covering SFT should forget less, and empirical evidence showing that RL forgets less. Second, the multi-modal analysis (Section 3.3) directly motivates why on-policy data (which induces mode-seeking behavior) preserves prior knowledge in multi-modal policies, providing the grounding for the practical findings in Section 4. That said, we agree the presentation can be improved to make this connection more explicit, and will revise the transition between Section 3 and Section 4 to clearly state: the KL perspective identifies on-policy sampling as the source of mode-seeking behavior, which Section 4 then empirically validates as the key factor.
>
>
>
> **Clarification on the contribution of Iterative-SFT.**
>
> > I view the algorithmic improvement in Sec 4 as some coarse-grained RL method with only positive learning. I think the contribution of this paper is not that enough. I am also wondering in what exact practical cases we need to use this new method.
>
> While there is a connection between Iterative-SFT and RL (as we acknowledge, citing RAFT and STaR), our contribution is not Iterative-SFT as a new algorithm but the insight that the degree of on-policyness determines forgetting. Iterative-SFT is part of the experimental design to test whether approximately on-policy data suffices. Without new algorithms, our finding is still actionable: practitioners who currently use SFT can substantially reduce forgetting by simply regenerating training data at each epoch, without switching to a full RL pipeline. This is directly useful for cases where reward models are unavailable or RL infrastructure is costly.
>
>
> **From Figure 7, it seems that the new method coverages slower than original SFT. I may misunderstand the figure, because the training curves of original SFT are not provided.**
>
> Thank you for raising this point. We believe there may be a misunderstanding of Figure 7. The SFT and Self-SFT curves appear as straight lines (not training curves) because they are trained for exactly 2 epochs and we plot only the final result, whereas Iterative-SFT is plotted across multiple rounds to show its trajectory. Iterative-SFT does require more rounds (up to 10) to reach the same target accuracy as SFT, which is the expected cost of regenerating data at each epoch rather than using a fixed off-policy dataset. However, the key comparison is at “matched target accuracy”: when Iterative-SFT reaches the same accuracy as SFT, it does so with dramatically less forgetting. For example, on IFEval with Qwen-2.5-7B-Instruct, Iterative-SFT reaches comparable accuracy to SFT (~60%) while its drop relative to GRPO remains near zero, compared to SFT's substantial drop of ~55 5%. We will add explicit per-step SFT training curves to Figure 7 in the revision so that convergence speed can be compared directly on the same axes.

---

> > ### Author Rebuttal · Reviewer_Dd4F · 2026-04-01
> >
> > The authors are clearly experts in the field, and both the manuscript and the rebuttal are highly professional. I have no further questions to raise. I will increase my scores, although I still find the conclusions somewhat less compelling than I had hoped.

---

> > > ### Author Response · Authors · 2026-04-07
> > >
> > > Thank you for your thoughtful feedback and for considering our rebuttal. We are glad our clarification addressed your concerns.

---

### Official Review · Reviewer_WByJ · 2026-03-12

**Soundness:** 2
**Presentation:** 2
**Significance:** 3
**Originality:** 2
**Overall Recommendation:** 3
**Confidence:** 4

**Summary:**

This paper presents a systematic empirical study of catastrophic forgetting in language-model post-training, with a focus on comparing supervised fine-tuning (SFT) and reinforcement learning (RL). Across multiple model families, model scales, and task types, the paper reports a consistent trend: RL achieves comparable or better target-task performance than SFT while inducing substantially less forgetting on non-target tasks. To explain this gap, the paper introduces a simplified KL-based analysis in which the model is viewed through a mixture perspective, and argues that the robustness of RL is primarily due to its reliance on on-policy data rather than other algorithmic ingredients such as KL regularization or advantage estimation. The paper further explores approximately on-policy variants and suggests that they can retain much of the forgetting-mitigation benefit at lower computational cost.

**Compliance With Llm Reviewing Policy:**

Affirmed.

**Final Justification:**

The authors' positive responses are helpful for readers. However, they may still have concerns about the verification of the experiments, which could also affect the reliability of the manuscript.

**Key Questions For Authors:**

The paper states that iterative SFT can show ''mild to no forgetting.'' Could the authors make this claim more quantitative? For example, can you report numerical thresholds, confidence intervals, or statistical comparisons against RL to clarify when the forgetting is negligible versus merely reduced?

Could the authors provide more detail on the SFT data-generation pipeline? In particular: (i) whether Llama-3.3-70B-Instruct may already have been exposed to the target tasks, (ii) the exact prompting strategy used to generate responses, and (iii) the precise filtering criteria applied before SFT training. These details seem important for assessing both fairness and reproducibility.

In Section 4.1, the comparison between REINFORCE and GRPO is used to argue that advantage estimation is not the main reason RL forgets less. However, REINFORCE also appears to underperform GRPO on the target task. Could the authors better control for this optimization gap, e.g., by matching target-task performance more closely, so that the forgetting comparison is less confounded by under-training?

The paper suggests that on-policy methods may help preserve safety behaviors. However, the experiments appear to test only whether safety-related capabilities are retained when adapting to non-safety tasks. Do the authors have evidence, or at least a more careful qualification, regarding whether the same conclusion should hold when safety alignment itself is the target task?

Section 3.2 presents the uni-modal analysis before clarifying that practical LMs are better thought of as multi-modal. Could the authors revise this part to more explicitly frame the uni-modal case as a theoretical baseline or counterexample, so readers do not over-interpret its practical relevance before reaching Section 3.3?

**Limitations:**

While the empirical study is substantial and the central finding is interesting, some of the stronger interpretations would benefit from tighter quantitative support and more careful qualification. In particular, the evidence for ''mild or no forgetting'' in approximately on-policy SFT is mostly qualitative, the REINFORCE-vs.-GRPO ablation does not fully isolate the role of advantage estimation from differences in optimization effectiveness, and the discussion of preserving safety behavior goes beyond the exact experimental setting studied in the paper. In addition, some important implementation details for evaluating fairness and reproducibility are not sufficiently explicit, especially regarding the teacher model used for SFT data generation and the data filtering pipeline. Finally, the simplified theoretical analysis is helpful as intuition, but its presentation could better distinguish between toy models and claims intended to explain practical language-model behavior.

**Strengths And Weaknesses:**

A major strength is the breadth of the empirical study: the comparison spans multiple model families, scales, and target tasks, and the main qualitative finding that RL forgets less than SFT appears consistent throughout the paper. The identification of on-policy data as the key explanatory factor is also useful for practitioners, since it yields an actionable takeaway beyond the narrower comparison between SFT and RL. In addition, the simplified theoretical analysis via KL direction and mixture modeling provides an intuitive lens for interpreting the empirical results, and the ablations on KL regularization, REINFORCE vs. GRPO, and approximately on-policy SFT make the overall argument more coherent. Overall, the manuscript studies a significant question and provides several interesting insights. That said, I also have a number of concerns. First, some claims are stated more strongly than the evidence currently supports. For example, the paper suggests that iterative or approximately on-policy SFT can exhibit ``mild to no forgetting,'' but the presentation would benefit from clearer quantitative thresholds or statistical comparisons against RL, rather than qualitative wording alone. Second, several implementation details important for fairness and reproducibility remain insufficiently specified, including whether Llama-3.3-70B-Instruct may have seen the target tasks during prior training, the exact prompting strategy used for generating SFT targets, and the filtering criteria applied before SFT training. Third, the ablation comparing REINFORCE and GRPO to argue against the role of advantage estimation is somewhat confounded by optimization quality: the paper itself notes that REINFORCE lags behind GRPO in target-task optimization, so similar forgetting levels may reflect under-training rather than equal robustness. Fourth, the discussion about preserving safety behaviors somewhat overreaches the presented evidence, since the experiments only test whether safety capabilities are retained while learning non-safety tasks, not whether on-policy methods preserve safety when safety alignment itself is the target of adaptation. Finally, the presentation of the simplified theoretical analysis could be improved: Section 3.2 initially presents the uni-modal case without clearly foregrounding that this is only a toy theoretical setting, while the relevance to practical LMs is deferred until the subsequent multi-modal discussion, which may temporarily confuse readers about the applicability of the result.

---

> ### Author Rebuttal · Authors · 2026-03-31
>
> Thank you for the detailed feedback and highlighting the breadth of our empirical study, insightfulness of our analysis, and the importance of identifying on-policy data as the primary contributor to RL’s robustness to forgetting. We treat your comments and questions below. If our response addresses your concerns, we would greatly appreciate it if you would consider raising your score. Please let us know if you have any further questions; we will gladly respond.
>
> **Making the claim "iterative SFT can show 'mild to no forgetting" more quantitative.**
>
> Thank you for the feedback. Note that Figure 7 (bottom row) already quantitatively shows that Iterative SFT (blue curve) exhibits a clear trend of lower drop compared to Self-SFT and SFT. The curve stays close to 0 throughout the multiple rounds of training. Nonetheless, in light of your comment, we will make textual description more precise and specify that the drop is within 2 points of accuracy.
>
> **Concerns about Llama-3.3-70B data contamination and the data generation pipeline.**
>
> We would like to first note that data contamination is a general concern for all academic work that relies on frontier open-weight models, and not a specific aspect of our experimental design. Moreover, we believe that any potential contamination only makes our results stronger. If Llama-3.3-70B-Instruct was exposed to the target task data, one would expect data to be less noisy because the teacher is more likely to generate the correct output, which should reduce forgetting during SFT. Yet our results show that SFT causes the model to forget significantly.
>
> For IFEval, MMLU, and Countdown, the exact prompts are provided in Appendix A.3. The teacher model generates responses using greedy decoding.
>
> For both SFT and Self-SFT, we keep only the correct responses using the same binary reward function used in RL. For Self-SFT, 5 responses are generated per prompt and all correct responses are retained. This is stated in Section 2.2, but in light of your comment we will make it more prominent in the revision to improve reproducibility.
>
>
> **Could the authors better control for this optimization gap for REINFORCE and GRPO?**
>
> Thank you for the helpful suggestion. We address this concern by roughly matching all three methods (SFT, GRPO, REINFORCE) at the same target-task gain and comparing forgetting.
>
> | | | IFEval | | MMLU | | Countdown | |
> |- |-|-|-|-|-|-|-|
> | | | Gain | Drop | Gain | Drop | Gain | Drop |
> | Llama 3.1 8B Inst. | SFT | 17.6 | 14.5 | 8.3 | 37.6 | 6.1 | 36.3 |
> | | REINFORCE | 17.7 | 7.8 | 8.3 | -0.1 | 7.5 | -0.7 |
> | | GRPO | 17.8 | 2.6 | 7.9 | -0.1 | 7.0 | -0.2 |
> | Qwen 2.5 7B Inst. | SFT | 3.7 | 6.0 | 6.2 | 19.6 | 10.3 | 29.2 |
> | | REINFORCE | 5.8 | 2.9 | 6.3 | -0.6 | 11.9 | -0.1 |
> | | GRPO | 6.6 | 0.6 | 7.3 | -0.2 | * | * |
>
> *: GRPO Countdown Qwen 7B: lowest available GRPO checkpoint (gain=24.0%, drop=1.2%) overshoots the matched gain, but notably still exhibits near-zero forgetting despite 2× higher gain than SFT.
> At matched target-task gain, SFT consistently forgets far more than both RL methods, directly ruling out the under-training confound. Both RL methods show similarly low forgetting at matched gain, confirming that the advantage estimator in GRPO primarily improves optimization rather than forgetting mitigation, and on-policy data remains the main factor.
>
>
> **The discussion about preserving safety behaviors somewhat overreaches the presented evidence.**
>
> The safety tasks are part of our broad non-target evaluation suite and not a focus of our claims. We include them because safety capabilities are particularly susceptible to degradation during fine-tuning (Qi et al., 2024), making them a suitable evaluation of forgetting.
> Regarding safety as the target of adaptation: one does not generally expect safety to degrade when directly training on safety data, as prior work consistently reports improvements (Dai et al., 2024). The relevant concern is whether other capabilities degrade, which is precisely what our paper studies. We will revise Section 7 to make this scope more precise.
>
> - Qi et al., 2024. Fine-tuning aligned language models compromises safety, even when users do not intend to! ICLR 2024.
> - Dai et al., 2024. Safe RLHF: Safe reinforcement learning from human feedback. ICLR 2024.
>
> **Clarification of the uni-modal setting as an analysis for helpful intuition.**
>
> Thank you for the suggestion. We do note this in the abstract, introduction, and preamble of Section 3, where we write that SFT can lead to less forgetting "if the initial policy has a single mode" but that "in multi-modal scenarios that arguably mirror more closely LM fine-tuning, [...] the mode-seeking properties of RL result in higher degrees of robustness to forgetting." That said, we will add a clear disclaimer at the beginning of Section 3.2 framing it as a theoretical baseline, and foregrounding that the practically relevant result follows in Section 3.3.

---

> > ### Author Rebuttal · Reviewer_WByJ · 2026-04-03
> >
> > Thank you for your explanation. However, I notice the impact of frontier models on data contamination. I wonder whether the authors might consider incorporating more recent models for validation, such as Qwen 3.5.  The authors do not perform validation in a multimodal environment, and I remain concerned about this limitation.

---

> > > ### Author Response · Authors · 2026-04-07
> > >
> > > Thank you for acknowledging our response; we appreciate your engagement in the discussion period.
> > >
> > > Regarding data contamination, it was not entirely clear to us whether our prior message fully addressed that concern. To clarify, the use of frontier teacher models such as Llama-3.3-70B-Instruct is standard in the literature, and we do not see a particular reason for it to be problematic in our case. This is especially true since, alongside SFT using labels from Llama-3.3-70B-Instruct, we also compare RL to self-SFT, where no teacher model is used.
> > >
> > > Due to the short time frame of the discussion period, we were unfortunately unable to provide additional results with a Qwen3.5 model or in environments with multiple modalities (such as text and images). We note that the Qwen3.5 model family was released after the ICML 2026 submission deadline, and that it is common for studies such as ours to focus on text-only settings. We therefore believe that the requested experiments, which were not raised in the original review, fall outside the scope of the current work. We sincerely hope that you can understand our perspective and thank you again for your time and effort.

---

### Official Review · Reviewer_vBLd · 2026-03-13

**Soundness:** 3
**Presentation:** 4
**Significance:** 4
**Originality:** 3
**Overall Recommendation:** 5
**Confidence:** 3

**Summary:**

This paper compares the susceptibility to catastrophic forgetting of SFT and RL during LLM post-training. Empirical evaluations and a simplified mixture-of-Gaussians model demonstrate that the mode-seeking nature of RL preserves multi-modal prior knowledge much better than the mode-covering nature of SFT.  Ablation studies rule out KL regularization and advantage estimation as primary causes and show that an efficient Iterative-SFT approach using approximately on-policy data can achieve high target performance while heavily mitigating forgetting.

**Compliance With Llm Reviewing Policy:**

Affirmed.

**Key Questions For Authors:**

Q1: The evaluation focuses heavily on tasks with exact or verifiable answers. Do the authors have any preliminary results or intuition on whether RL's robustness to forgetting holds for more open-ended generative tasks, such as long-form summarization, multilinguality, or complex coding tasks?

Q2: Could the authors provide a quantitative breakdown of the computational costs comparing Iterative-SFT, full GRPO, and standard SFT?

Q3: How do the authors explain the notable exception for Llama models on the IFEval task shown in Figure 6, where KL regularization clearly impacts the gain-drop tradeoff? Doesn't this specific result suggest that the importance of KL regularization in mitigating forgetting might actually be highly model- and task-dependent, rather than universally negligible?

**Limitations:**

Yes

**Strengths And Weaknesses:**

### Strengths

1. The paper introduces an intuitive, simplified theoretical explanation using uni-modal vs. multi-modal Gaussian distributions.

2. This paper isolates the specific mechanism behind RL's robustness, showing non-regularized GRPO and standard REINFORCE still mitigate forgetting.

3. The proposed Iterative-SFT is a highly valuable solution, offering a computationally lighter alternative to full RL while inheriting the forgetting-mitigation benefits of on-policy data sampling.

### Weaknesses

1. Mode-seeking (RL) can sometimes lead to severe mode collapse (e.g., repeating the exact same phrase to hack a reward). The paper does not empirically measure mode collapse or diversity loss on the target task to see if RL's retention comes at the cost of generation diversity.

2. The non-target tasks are restricted to math, safety, and multiple-choice benchmarks. It remains unproven if RL's mode-seeking nature preserves broader, open-ended capabilities like summarization, multilinguality, or coding.

3. The paper states that Iterative-SFT has a lower computational cost than RL without a quantitative analysis of FLOPs, wall-clock time, or memory footprint comparing Iterative-SFT, full GRPO, and standard SFT to justify the efficiency claim.

4. The paper broadly claims KL regularization is not a major factor in mitigating forgetting, yet Figure 6 shows a clear exception where it significantly degrades performance for Llama on IFEval. Dismissing this practical exception makes the conclusion overstated.

---

> ### Author Rebuttal · Authors · 2026-03-31
>
> Thank you for the insightful feedback and support. We address your comments below. Please let us know if you have any further questions, we will gladly elaborate.
>
> **The paper does not empirically measure mode collapse or diversity loss on the target task to see if RL's retention comes at the cost of generation diversity.**
>
> Thank you for raising this matter. We note that in Appendix A.6 (Table 3), we evaluate RL-trained models on AlpacaEval, an open-ended conversational benchmark where mode collapse would manifest as degraded or repetitive outputs judged by GPT-4. Across both Llama-3.1-8B-Instruct and Qwen-2.5-7B-Instruct, RL preserves or even slightly improves AlpacaEval win rates compared to the initial policy, suggesting that generation diversity is not severely compromised. We will clarify this point in the main text. Furthermore, we are happy to add explicit diversity measurements in the revision to directly quantify this, and would welcome any specific metrics the reviewer has in mind
>
>
> **The non-target tasks are restricted to math, safety, and multiple-choice benchmarks. It remains unproven if RL's mode-seeking nature preserves broader, open-ended capabilities like summarization, multilinguality, or coding.**
>
> The AlpacaEval experiments mentioned above (Appendix A.6; Table 3) indicate that RL preserves broader, open-ended capabilities beyond the verifiable tasks in our main experiments. In particular, the results are consistent with our main findings: for both Llama-3.1-8B-Instruct and Qwen-2.5-7B-Instruct, SFT substantially degrades AlpacaEval win rate (e.g., from 41.6% to 0.0% after training on Countdown), whereas RL maintains or even slightly improves performance. Nonetheless, AlpacaEval does not cover every open-ended domain. We therefore believe that considering additional domains is a valuable direction, yet one that falls outside the scope of the current work.
>
> **Could the authors provide a quantitative breakdown of the computational costs comparing Iterative-SFT, full GRPO, and standard SFT?**
>
> We provide a FLOPs analysis for Qwen 2.5 7B Instruct on IFEval. Let $P$ denote model parameters ($\sim$ 7.6B), $N$ training prompts ($\sim$ 10K), $L$ average sequence length ($\sim$ 1500 tokens), $L_g$ average generation length (~1000 tokens), $G=5$ generations per prompt, and $B=128$ batch size. We use standard approximations of $6PL$ FLOPs per training example (forward + backward) and $2PL_g$ per generated sequence.
>
> **SFT:** generates $G$ responses per prompt once upfront, filters for correctness ($N_{filt} \approx 3K$), then trains $E=2$ epochs.
>
> $$FLOPs_{SFT} = N \cdot G \cdot 2PL_g + 6PL \cdot N_{filt} \cdot E \approx 1.9 \times 10^{18}$$
>
> **GRPO:** trains $E=2$ epochs ($T \approx 156$ steps); at each step, generates $B \times G$ responses, performs forward+backward on all prompt+response sequences, plus a reference model forward for KL.
>
> $$FLOPs_{GRPO} = T \cdot B \cdot G \cdot (2PL_g + 6PL + 2PL) \approx 1.1 \times 10^{19}$$
>
> **Iterative-SFT:** runs $R \approx 7$ rounds on average (ranging from 3 to 10 across experiments); each round generates $N \times G$ responses, filters for correctness, and trains 1 epoch on the filtered subset.
>
> $$FLOPs_{Iter\text{-}SFT} = R \cdot (N \cdot G \cdot 2PL_g + 6PL \cdot N_{filt}) \approx 6.8 \times 10^{18}$$
>
> | Method | FLOPs | Relative to SFT |
> |---|---|---|
> | SFT | $1.9 \times 10^{18}$ | 1x |
> | Iterative-SFT | $6.8 \times 10^{18}$ | ~4x |
> | GRPO | $1.1 \times 10^{19}$ | ~6x |
>
> Iterative-SFT requires $\sim$ 1.6x fewer FLOPs than GRPO. In practice, the wall-clock gap is even larger: GRPO requires synchronous model weight syncing between training and generation workers at every step ($\sim$ 156 syncs), whereas Iterative-SFT decouples generation and training into sequential phases, syncing only once per round ($\sim$ 7 syncs). GRPO also requires loading a reference model for KL computation, which Iterative-SFT avoids entirely. We will include this analysis in the revision.
>
> **Clarification on the claim on the KL regularization not being a major factor in mitigating forgetting.**
>
> > Figure 6 shows a clear exception where it significantly degrades performance for Llama on IFEval. Dismissing this practical exception makes the conclusion overstated.
>
> Thank you for raising the point. In the Llama trained on IFEval case, while removing KL regularization increases the drop from 1.6% to 10.2% (Llama 1B) and from 3.4% to 11.7% (Llama 8B), these drops remain substantially smaller than the corresponding SFT drops (26.2% for Llama 1B and 27.8% for Llama 8B). This suggests that on-policy data still provides the dominant forgetting mitigation, while KL regularization can provide additional benefit in a model- and task-dependent manner. We will revise the text to explicitly acknowledge this exception and soften the claim to: "KL regularization is not the primary contributor, though it can provide meaningful additional benefit in certain model–task combinations."

---

> > ### Author Rebuttal · Reviewer_vBLd · 2026-04-04
> >
> > Thanks for your detailed response, and I will maintain my positive evaluation.

---

> > > ### Author Response · Authors · 2026-04-07
> > >
> > > Thank you for your time in reviewing our rebuttal and for the positive feedback. We are glad that the additional information adequately answered your questions.

---

### Decision · Program_Chairs · 2026-04-30

**Decision:**

Accept (regular)

**Comment:**

This submission provides a systematic empirical investigation into catastrophic forgetting during the post-training of large language models, specifically comparing Supervised Fine-Tuning (SFT) and Reinforcement Learning (RL). The authors identify on-policy data sampling as the primary mechanism for RL's superior retention of prior knowledge,

The observation that RL mitigates forgetting is not entirely new and has been published recently [1, 2, 3]; however, the authors' mechanistic identification of on-policy data as the cause is a distinct contribution. The study is rigorous, spanning multiple model families (Llama, Qwen) and scales. The authors also introduce "Iterative-SFT" as a practical middle ground, demonstrating that practitioners can mitigate forgetting by moving toward on-policy data without the full complexity of an RL pipeline.

Initial concerned raised by some reviewers regarding memory footprint and computational complexity (andf wall-clock time) have been addressed by the authors during rebuttal.

Overall, the paper presents a solid, well-executed study on a timely and important topic. It provides both theoretical intuition and practical evidence that will be highly relevant to the ICML community and confirms recent 3rd party work. Despite some initial concerns regarding the novelty of the empirical observations, the authors' focus on the underlying mechanisms—and their thorough rebuttal—justifies acceptance.



[1] RL's Razor: Why Online Reinforcement Learning Forgets Less (Shenfield et.al, 2026)
[2] RL Is Neither a Panacea Nor a Mirage: Understanding Supervised vs. Reinforcement Learning Fine-Tuning for LLMs
[2] Mitigating Forgetting Between Supervised and Reinforcement Learning Yields Stronger Reasoner